# Testing Two Somatic Cell Count Cutoff Values for Bovine Subclinical Mastitis Detection Based on Milk Microbiota and Peripheral Blood Leukocyte Transcriptome Profile

**DOI:** 10.3390/ani12131694

**Published:** 2022-06-30

**Authors:** Jinning Zhang, Wenlong Li, Yongjie Tang, Xueqin Liu, Hailiang Zhang, Yueling Zhou, Yachun Wang, Wei Xiao, Ying Yu

**Affiliations:** 1Key Laboratory of Animal Genetics, Breeding and Reproduction, Ministry of Agriculture and Rural Affairs & National Engineering Laboratory for Animal Breeding, College of Animal Science and Technology, China Agricultural University, Beijing 100193, China; s20203040560@cau.edu.cn (J.Z.); s20183040510@cau.edu.cn (W.L.); cautyj@163.com (Y.T.); liuxueqin@cau.edu.cn (X.L.); zhl108@cau.edu.cn (H.Z.); zhouyueling@cau.edu.cn (Y.Z.); wangyachun@cau.edu.cn (Y.W.); 2Beijing Animal Husbandry Station, Beijing 100029, China

**Keywords:** dairy cows, somatic cell count, milk microbiota, subclinical mastitis, transcriptome

## Abstract

**Simple Summary:**

Setting a proper SCC cut-off value for determining the intramammary infection status of each individual cow is essential to early mastitis detection and prevention. In this study, we compared the effect of two commonly used SCC thresholds on distinguishing milk microbiota and host gene expression patterns, and demonstrated that the microbial composition and peripheral blood leukocyte transcriptome profiles did have conspicuous differences between cows with SCC above and below 100,000 cells/mL, respectively, which may help establish why 100,000 cells/mL is a more suitable cow-level cut-off value of subclinical mastitis diagnosis than 200,000 cells/mL from the perspective of microbiota and transcriptomic responses.

**Abstract:**

Somatic cell count (SCC) is an important indicator of the health state of bovine udders. However, the exact cut-off value used for differentiating the cows with healthy quarters from the cows with subclinical mastitis remains controversial. Here, we collected composite milk (milk from four udder quarters) and peripheral blood samples from individual cows in two different dairy farms and used 16S rRNA gene sequencing combined with RNA-seq to explore the differences in the milk microbial composition and transcriptome of cows with three different SCC levels (LSCC: <100,000 cells/mL, MSCC: 100,000–200,000 cells/mL, HSCC: >200,000 cells/mL). Results showed that the milk microbial profiles and gene expression profiles of samples derived from cows in the MSCC group were indeed relatively easily discriminated from those from cows in the LSCC group. Discriminative analysis also uncovered some differentially abundant microbiota at the genus level, such as *Bifidobacterium* and *Lachnospiraceae_AC2044_group*, which were more abundant in milk samples from cows with SCC below 100,000 cells/mL. As for the transcriptome profiling, 79 differentially expressed genes (DEGs) were found to have the same direction of regulation in two sites, and functional analyses also showed that biological processes involved in inflammatory responses were more active in MSCC and HSCC cows. Overall, these results showed a similarity between the milk microbiota and gene expression profiles of MSCC and HSCC cows, which presented further evidence that 100,000 cells/ml is a more optimal cut-off value than 200,000 cells/mL for intramammary infection detection at the cow level.

## 1. Introduction

Subclinical mastitis is known as one of the most prevalent mammary diseases plaguing the dairy industry. It can have adverse effects on the milk quality and cause economic losses. Early diagnosis of bovine subclinical mastitis is difficult, as there are no clinical signs. Fortunately, the association between abnormal somatic cell counts and the incidence of clinical or subclinical mastitis has been identified and for now, somatic cell count monitoring is still used as an effective method for subclinical mastitis detection [1,2]. Therefore, finding a reliable threshold for delineation of subclinical mastitis is essential to the prevention of this disease. However, the exact cut-off value has remained quite controversial. In some countries, 100,000 cells/mL was regarded as a quarter-level threshold for differentiating mammary glands with or without infection [3,4,5]. Several studies have confirmed that using 100,000 cells/mL might be more appropriate. In a study carried out by Forsback et al. [6], the result revealed that for each individual cow, milk from the udder quarters with SCCs over 100,000 cells/mL underwent changes in composition, such as lower casein number and content of lactose compared with healthy quarters (SCCs < 100,000 cells/mL). By constructing a linear mixed model to quantify the relationship between milk loss and somatic cell counts, Hand et al. [7] found acute lactation milk losses occurred in lactations with 5 or more 24-h test days where SCC exceeded 100,000 cells/mL. Despite the widespread use of this value in differentiating subclinical mastitis udders from healthy ones, some other thresholds were being used or proposed, among which the most extensively used cut-off value is 200,000 cells/mL [8,9]. For now, composite samples are less well studied than quarter milk samples, but because of the possible failure of bacteria isolation from quarter samples due to low bacteria concentration [9], we used composite milk samples as a proxy to compare the two SCC thresholds for identifying individual cows at risk of intramammary infection despite the dilution effect and low sensitivity of the SCC test using composite samples.

To study the microbial composition shift caused by diseases and to help us understand the roles of the microbiota in the development of some bacteria-caused diseases, 16S rRNA gene sequencing was intensively utilized. Compared with traditionally used quantitative PCR (qPCR) for mastitis pathogen detection, 16S rRNA can almost characterize the entire bacterial profile, which is more conducive to studying subclinical mastitis, because research has indicated that commensal and potential beneficial bacteria are also involved in the subclinical mastitis incidence rate [10]. By using this technique, several studies have shown distinct microbial profiles between milk samples from healthy cows and pathogen infected ones. Wang demonstrated that the composition of microbial communities in milk samples derived from healthy, subclinical mastitis and mastitis cows had a significant difference, with the abundance of common pathogens such as *Klebsiella* and *Streptococcus* higher in cows suffering from subclinical mastitis [1,10]. Apart from these pathogens, an increase in certain previously unconcerned bacteria, such as *Cutibacterium*, *Colidextribacter*, and *Paeniclostridium* has also been shown to be correlated with adverse udder health [11], suggesting 16S amplicon sequencing can more comprehensively reflect the constitution of the bacterial community—including both commensal and pathogenic bacteria—than qPCR.

Global functional genomics technologies enable us to explore the dynamic changes in response to the environmental stimulus and underlying diseases. RNA-seq, which can simultaneously quantify the expression of a large number of genes, is quite suited for investigation of the complex host–pathogen interplay occurring during mastitis [12]. Significant changes in the leukocyte transcriptome attributed to subclinical or clinical mastitis have been reported and the degree of the changes was positively correlated with the severity of the infection [13].

As indicated, both milk microbiota and the host transcriptional profile can help investigations of the differences between healthy and infected cows. Therefore, the objectives of this study were to utilize RNA-seq and 16S rRNA gene sequencing to determine whether using 100,000 cells/mL as an SCC cut-off value for intramammary infection diagnosis is better than using 200,000 cells/mL at the cow level and explore the underlying reason from the perspective of the bacterial community and global gene expression. 

## 2. Materials and Methods

### 2.1. Sample Collection

In this study, sample collection was conducted in two well-managed commercial dairy farms located in two different cities (site1: Beijing, site2: Hangzhou) in China and in two different seasons. To minimize the effect of recent mammary infections, all cows (Holstein cows) selected in this study were free of clinical mastitis and antibiotic treatment for at least 2 consecutive months before the sample collection. Milk samples were collected aseptically, following cleaning of teat ends with 70% ethanol. Milk from each of the 4 quarters was evenly mixed in a 100 mL-sterile tube after discarding the first few streams of milk. Samples were kept on dry ice during the whole collection and transfer process and were later stored at −80 °C until DNA extraction. A total of 121 milk samples were divided into the following three groups based on SCC value: low SCC group (LSCC): <100,000 cells/mL; medium SCC group (MSCC): 100,000–200,000 cells/mL; high SCC group (HSCC): > 200,000 cells/mL. A blood sample was aseptically taken from the caudal vein of each cow and stored in a 10-mL pre-added EDTA-K2 vacutainer on the same day the milk samples were collected.

### 2.2. S rRNA Gene Sequencing Procedure and Bioinformatic Analysis

Total genome DNA was extracted from milk using the cetyltrimethylammonium bromide (CTAB) method. The purity of DNA was assessed by 1% agarose gel electrophoresis. The V3-V4 hypervariable region of bacterial 16S rRNA genes was then amplified using universal primers 341F (5′-3′: CCTAYGGGRBGCASCAG) and 806R (5′-3′: GGACTACNNGGGTATCTAAT) with the barcode to tag the PCR products from each sample. PCR products were mixed in equidensity ratios to generate a mixture which was then purified with the Qiagen Gel Extraction Kit (Qiagen, Germany). The purity and integrity of the PCR products were evaluated using Agilent5400 (Agilent, Santa Clara, CA, USA). Sequencing libraries were generated using the TruSeq^®^ DNA PCR-Free Sample Preparation Kit (Illumina, San Diego, CA, USA) following the manufacturer’s recommendations. Amplification was performed under the following conditions: denaturation at 98 °C for 1 min, then 30 cycles of denaturation at 98 °C for 10 s, followed by annealing at 50 °C for 30 s, elongation at 72 °C for 30 s and a final step at 72 °C for 5 min. Library quality was assessed on a Qubit@ 2.0 Fluorometer (Thermo Scientific, Waltham, MA, USA) and Agilent Bioanalyzer 2100 system. Qualified library was later sequenced on Illumina NovaSeq 6000 platform and generated 250 bp paired-end reads.

Microbiota profiling was carried out following Qiime2 (v 2020.8) pipeline. Amplicon sequences were assigned to each sample based on the tagged barcode. Quality control was accomplished by cutting off the adaptors and using Divisive Amplicon Denoising Algorithm (DADA2) to correct sequence errors, filter chimeric sequences, and merge the paired-end reads with a minimum overlap of 30 nt. High-quality reads that survived from the filtration were summarized to generate an amplicon sequence variant (ASV) table. A Naïve Bayes classifier (a plugin in Qiime2) was trained based on sequences extracted from the SILVA database release 138 and was then applied to ASV to perform taxonomic classification. Contaminating mitochondrial and chloroplast sequences were removed from the feature table. 

Downstream microbiota analyses were conducted using the R package vegan. Rarefaction curves of richness were generated at the genus level to test the adequacy of the sequencing depth. A total of 5 α indices (Chao1 richness, Shannon, Simpson, Inverse Simpson, ACE) were calculated to estimate alpha diversity. Pairwise comparisons of the alpha diversity indices were conducted by constructing a linear mixed model to correct site effect, lactation stage, and parity using the R package lmerTest (v 3.1.3) and emmeans. The parity and lactation stage were classified into two levels (primiparous and multiparous for parity, middle (35–100 d) and late (>100 d) for lactation stage) as research suggested that parity and stage of lactation may have an effect on overall diversity of the udder microbiota [14]. The site was dichotomized as 0 or 1 and was used as a covariate for correction. It should be noted that the site and season of collection are confounded due to the fact that samples were collected from two different dairy farms in two different seasons. For brevity, this confounded effect was referred to as the ‘site effect’ instead of the ‘site/season effect’ throughout the manuscript. Principal coordinates analysis (PCoA) based on Bray–Curtis distance matrices was used to show dissimilarity in microbial composition (β-diversity). In order to examine the effect of various experimental factors on the variation of microbial communities among samples, we performed a distance-based redundancy analysis test using the multivariate model which incorporated the same factors as above. Multicollinearity of variables was examined using variation inflation factors (VIF), and a value less than 10 means no multicollinearity. Significance for each factor was determined using 9999 permutations. Significant factors were then chosen using the forward selection procedure with 9999 permutation tests. For investigation of underlying differences in metagenome functions among the three groups, Phylogenetic Investigation of Communities by Reconstruction of Unobserved States (PICRUSt2) was used for prediction of metabolic pathway abundances. To infer the relationship between microbes, an individual co-occurrence network was constructed based on Spearman correlation between genera via R package psych. Network parameters reflecting the topological properties of the network were calculated using R package igraph (v 1.2.11). Differentially abundant genera and metabolic pathways were identified using the generalized linear model (glm) in ALDEx2 based on Monte Carlo sampling from the Dirichlet distribution for each sample. Other variables, such as parity and site, were also taken into account when performing glm analysis.

### 2.3. RNA-Seq and Transcriptomic Data Processing

Total RNA was isolated from peripheral white blood cells using Trizol reagent (Ambion, Austin, TX, USA) following the manufacturer’s instructions. Quality and integrity check were performed using Nanodrop (GE, USA) and Aglient 2100 BioAnalyzer (Agilent, USA), respectively. Sequencing was conducted on the Illumina NovaSeq 6000 platform (Illumina, USA) and generated 150 bp paired-end reads. Adapters and reads with low quality and a length less than 36bp were removed by Trimmomatic (v. 0.39) before the alignment to the bovine reference genome ARS-UCD1.2 using STAR (v. 2.7.9a). A total of 36 transcriptomic datasets were included in this study (18 per site and 6 per group). Reads mapped to the reference genome were quantified by feature Counts in subread (v. 2.0.2) package. Genes with low expression levels (counts per million < 1) were removed from the downstream analysis. Differentially expressed gene (DEG) analysis was performed separately for samples collected from different sites using R package DESeq2 with the covariates mentioned before being corrected. Read count normalization was accomplished using rlog transformation. Functional analyses including KEGG and Gene Ontology (GO) enrichment analysis were carried out using the R package Clusterprofiler (v 3.18.1). To merge the transcriptomic data from the two sites to conduct the following analysis, the batch effect was removed using the R package limma (v 3.46.0). To test for the associations between differentially abundant taxa and top differentially expressed genes, the R package MaAsLin was used to assess the relationship between DEG and microbiota with the following model: microbial abundance~top DEGs + site + parity + lactation stage. Significance was determined at *p* < 0.05. Additionally, Gene Set Variation Analysis (GSVA) was conducted to confirm the result of enrichment analysis and further estimate changes in the activity of biological process (BP)-related pathways.

## 3. Results

### 3.1. Taxonomic Profiling Analysis and Core Microbiota Identification

The rarefaction curves of all samples reached a plateau, suggesting a sufficient amount of sequencing data (Appendix A). Four samples with the number of observed genera three standard deviations (3SD) below the mean value were removed from the downstream analysis. For comparison purposes, 87 samples were randomly selected (29 per group) for downstream analysis, making samples more evenly distributed among groups with different SCC levels.

Taxonomic annotation identified 34 phyla and 755 genera in total from the 87 composite milk samples. As displayed in Figure 1a, the top 5 dominant phyla were *Firmicutes* (0.42 ± 0.14), *Bacteroidota* (0.21 ± 0.07), *Actinobacteriota* (0.20 ± 0.08), *Proteobacteria* (0.08 ± 0.08), and *Patescibacteria* (0.05 ± 0.05) and the top 10 most abundant genera were *Oscillospiraceae_UCG-005*, *Corynebacterium*, *Bacteroides*, *Ornithinimicrobium*, *Christensenellaceae_R-7_group*, *Oscillospirales_UCG-010*, *Paracoccus*, *Romboutsia*, *Rikenellaceae_RC9_gut_group,* and *TM7a*, constituting more than 33% of the total relative abundance (Figure 1b). Notably, phylum *Patescibacteria* showed a site-specific pattern with the relative abundance in samples from site2 (one farm in Hangzhou, China) significantly higher than those from site1 (one farm in Beijing, China), suggesting the core phyla of the milk microbiota may vary from site to site.

### 3.2. Pairwise Comparison of Microbial Community Diversity of Groups with Different SCC Levels

To evaluate the difference in alpha diversity of milk microbial composition in LSCC, MSCC, and HSCC groups, we chose the linear mixed model with the smallest Akaike information criterion (AIC) to correct covariates that might affect the result. Unfortunately, there was no significant difference in alpha diversity among the three groups using any of the five alpha indices after correction for cofounding factors (Appendix A). The adjusted alpha diversity indices showed that the microbial diversity of LSCC samples was slightly lower than that of MSCC and HSCC groups (Figure 2). Similarly, PCoA did not show a clear separation among samples in different groups, suggesting a negligible difference in the overall microbial composition at the genus level. However, samples collected from different sites showed distinct clustering patterns and explained much of the variation (PCoA1: 40%) (Figure 3). To statistically interpret the result, pairwise distance-based redundancy analysis at the genus level was conducted to assess the effect of each factor on beta diversity by taking samples from two out of the three groups. Alternatively, we also merged LSCC and MSCC into one single group (LMSCC) and HSCC and MSCC into HMSCC to see the impact of different SCC cut-off values on microbiome variation. As shown in Table 1, site was the most significant factor affecting the microbial composition, regardless of the groups used for the analysis. 

Between-group composition variation was significant for analysis of milk microbiota by group and was performed using groups LSCC and HSCC and groups LSCC and MSCC. By merging LSCC and MSCC into one single group (LMSCC), the composition between LMSCC and HSCC did not significantly differ (*p* = 0.41). Conversely, significant difference in the composition existed between the HMSCC and LSCC groups (*p* < 0.02).

A stepwise forward selection procedure was performed for each model that incorporated different pairs of groups. The result further confirmed that site had the greatest effect on milk microbiota composition and explained over 20 percent of variation in all models (R^2^: 0.20–0.27, *p* = 10^−4^). Group factor was selected as a dependent explanatory variable when analysis was conducted using (1) LSCC and MSCC groups; (2) LSCC and HSCC groups; (3) HMSCC and LSCC groups. Together, these results demonstrated that the microbial composition of MSCC shared more similarity with that of HSCC, and using a cut-off value of 100,000 cells/mL could discriminate the variation in microbial composition.

### 3.3. Identification of Differentially Abundant Genera and Metabolic Pathways in Groups with Different SCC Levels

To identify the most relevant bacteria in each group, we performed pairwise comparative analysis between any two groups at the genus level. As before, we also merged LSCC and MSCC or HSCC and MSCC into one group to see which cut-off value could yield more differentially abundant genera. Only bacteria taxa present in at least 20% of the samples used in comparisons were kept for the differential abundance analysis. A total of 33 differentially abundant genera were identified in LSCC vs. MSCC comparison (*p* < 0.05), among which 27 were observed to be more prevalent in LSCC and 6 in MSCC. For HSCC vs. MSCC comparison, only one differentially abundant genus (*Planktosalinus*) was detected and had higher abundance in MSCC. 

The result also showed that more differentially abundant genera existed in LSCC vs. MSCC comparison than in HSCC vs. MSCC comparison, and in HMSCC vs. LSCC comparison (*n* = 29) than in LMSCC vs. HSCC comparison (*n* = 3), indicating that the microbial signature of MSCC cows was more different from that of LSCC than that of HSCC cows (Figure 4a). Notably, two genera (*Bifidobacterium* and *Lachnospiraceae_AC2044_group*) (*p*_adj_ < 0.05) that were most significantly differentially abundant in LSCC vs. MSCC comparison were also identified as differentially abundant genera in LSCC vs. HSCC comparison. In addition, the relative abundance of these two genera was higher in the group with lower SCC (Figure 4b) and site had little effect on the difference in the abundance (*p* > 0.05). 

To investigate the differences in the metagenome function contents among the three groups, we used predicted metabolic pathways with prevalence of no less than 30% for comparison analysis. Using the same method as differentially abundant genera identification, we found that three metabolic pathways (Formylmethanofuran--tetrahydromethanopterin *N*-formyltransferase, Nicotinamide phosphoribosyltransferase, Methenyltetrahydromethanopterin cyclohydrolase) were shared in all comparisons and more abundant in the group with a lower SCC level (Figure 4d). Consistent with the result of differential abundant genera analysis, we also found that the number of significantly differential metabolic pathways was larger in LSCC vs. MSCC (*n* = 104) than in MSCC vs. HSCC (*n* = 57), and in HMSCC vs. LSCC comparison (*n* = 29) than in LMSCC vs. HSCC comparison (*n* = 3). 

### 3.4. Comparative Analysis of Microbial Interaction Network

Microbial networks were constructed separately in samples derived from each of the three groups in each of the two sites. To avoid potential bias, we randomly selected samples of the same number for each group based on the minimum group size (site1: 7/per group, site2: 13/per group) and only kept the top 100 most abundant genera with prevalence > 60%. Nodes with a significant correlation > 0.8 were kept for network visualization (Figure 5). Network topological features such as degree (number of edges connected to a code), betweenness centrality (a node’s importance as a hub in the network), closeness centrality (inverse of the distance from the node to all other nodes), and transitivity (clustering coefficient) were used for comparison (Table 2).

A Kolmogorov–Smirnov test of these parameters showed that the closeness centrality was almost significantly different between any two given groups regardless of the sites, suggesting that the central position of each genus in the network was distinct among the three groups. Interestingly, we also found the closeness centrality of differentially abundant genera had large differences between LSCC and MSCC, and the tendency showed consistency in both sites (R_adj_^2^= 0.18, *p* < 0.001) (Appendix A).

### 3.5. Identification of Differentially Expressed Genes between Groups

Figure 6a,c showed the number of DEGs identified in each group comparison in each site. Comparison of LSCC and HSCC obtained the greatest number of DEGs (1068 for site1 and 455 for site2) using statistical significance *p* < 0.05 and |log2(FoldChange)| ≥ 1. In samples from both sites, more DEGs were identified in LSCC vs. MSCC (363 for site1 and 955 for site2) than in HSCC vs. MSCC (63 for site1 and 270 for site2). Furthermore, cluster analysis and expression heatmap of the most significant DEGs (FDR < 0.1) in both sites revealed apparently distinct transcriptional profiles between LSCC and the other two groups (Figure 6b,d), indicating that the transcriptional profile of MSCC cows was more different from that of LSCC compared to HSCC cows. A total of 7, 32, and 72 DEGs were shared in HSCC vs. MSCC comparison, LSCC vs. MSCC comparison and LSCC vs. HSCC comparison in both sites, respectively. Among those DEGs, 70 DEGs had the same regulation direction in both sites. Furthermore, the correlation between the expression profile of these common DEGs in the two sites was significant (R_adj_
^2 ^ =  0.9, *p* < 0.0001) (Appendix A). Unfortunately, among these common DEGs, no genes were found to be shared in LSCC vs. MSCC and LSCC vs. HSCC comparison.

Later, KEGG pathway and GO enrichment analyses were performed to investigate the functions of these common genes. We found KEGG pathways that were most enriched by these genes were cytokine–cytokine receptor interaction, NF-κ B signaling pathway, viral protein interaction with cytokine and cytokine receptor, and apoptosis (FDR < 0.1), suggesting that these genes play an important role in inflammatory response. Significant enriched biological processes in GO terms (*p* < 0.05) were also mainly involved in the cytokine-mediated signaling pathway and positive regulation of inflammatory response. In addition, GSVA further confirmed immune-related biological processes, such as regulation of interleukin 6 mediated signaling pathway, B cell chemotaxis, and positive regulation of T cell apoptotic process, were lowly expressed in LSCC, compared with the other two groups (Figure 7a–c).

### 3.6. Correlation Analysis between Microbiota and Gene Profiles

To understand the association between the top discriminating genes and the differentially abundant genera, a mixed linear model controlling site effect, parity, and lactation stage was constructed to fit for all common DEGs with the same regulation direction and FDR < 0.1 in both sites and each of the differentially abundant genera identified in each group comparison with prevalence >0.5. Overall, 33 significant associations were detected (*p* < 0.05) (Figure 8a). Among these associations, a strong negative association was detected between the gene *S100A9* upregulated in HSCC and bacteria *Prevotellaceae_UCG-003* and *Muribaculaceae,* which were proportionally higher in LSCC, and another pair of LSCC prevalent bacteria *UCG-005* and HSCC upregulated gene *GADD45G.* The expression levels of both genes were highest in HSCC and lowest in LSCC, corresponding with the SCC level (Figure 8b).

## 4. Discussion

Early diagnosis of subclinical mastitis is crucial for therapeutic decisions and for now, SCC is still broadly used as a proxy for subclinical mastitis [15]. To our knowledge, this is the first study to explore the appropriate threshold for subclinical mastitis or intramammary infection diagnosis from the perspective of both milk microbiota and peripheral blood leukocyte transcriptome, which are novel high-throughput intermediate molecular phenotypes in recent years. 

Though the mammary gland is the organ directly involved in the synthesis and secretion of milk, it is not always the most ideal material to study intramammary diseases due to collection difficulty and effect on the lactating cows. Blood, however, is relatively easy to collect and is responsible for various substance and molecule exchanges with the mammary gland. Moreover, the presence of some immune cells within the blood, such as leukocytes, enables blood transcriptome to reflect the overall physiological condition of cows [16,17]. Therefore, our study used peripheral blood leukocyte transcriptome as a proxy for mammary gland transcriptome. Yet this may raise another problem as other undergoing inflammatory reactions may also contribute to the changes in the expression levels of immune response-associated genes, thus biasing the result. We believe this limitation was addressed through the large sample size and comparison of samples from two different dairy farms.

The present study identified an enrichment of genera *Bifidobacterium* and *Lachnospiraceae_AC2044_group* in milk with SCC < 100,000 cells/mL. Previous studies have referred to *Bifidobacterium* as one of the markers of “healthy microbiota” and shown that it has lower relative abundance in cows with udder pathology [18]. Additionally, intramammary infusion of *Bifidobacterium breve* was reported to have an effect on clearance of minor mastitis causing pathogens from infected quarters [19]. Genera *Lachnospiraceae_AC2044_group* has not been identified as a marked beneficial bacterium in uninfected mammary quarters before. However, a higher percentage of unclassified *Lachnospiraceae* was detected in milk samples derived from healthy quarters in one study [20]. 

For the first time, the topological properties of the co-occurrence networks in milk microbiota were compared among groups with different SCC levels to illustrate how microbial interactions change with the increase in SCC. Closeness centrality represents the tightness of the network, and a higher closeness centrality usually indicates the greater importance of the node [21]. While both sites had almost the same tendency in which the closeness centrality of differentially abundant genera was lower in LSCC compared with MSCC, the closeness centrality in HSCC was not consistent between two sites, possibly because the co-occurrence network might become more unstable with the increase in the severity of the infection or the disease [22,23].

Samples from two different farms were used in this study to provide a result that may have a wider application. However, correcting batch effects for microbiota studies remains a problem because the same methods that work well for other types of omics data are not necessarily appropriate for microbiome datasets, which are extremely sensitive to the variation in experimental and computational processing [24]. The ALDEx2 was selected in this study for differential abundance testing because of its high performance in producing consistent results across studies and agreeing with the intersection of results from different approaches [25]. 

In this study, the correlation between the top DEGs and differentially abundant genera were estimated to help understand their roles in host–microbe interplay. S100 calcium-binding protein A9 encoding gene *S100A9* has previously been shown to be actively involved in inflammatory processes and upregulated in bovine mastitis [26,27]. Furthermore, it was enriched in some inflammatory response-related biological processes in this study, such as neutrophil chemotaxis, positive regulation of intrinsic apoptotic signaling pathway, and leukocyte cell–cell adhesion. *Muribaculaceae,* a beneficial commensal bacteria demonstrated by several studies, had a lower relative abundance in cows with subclinical mastitis [28,29] and was negatively correlated with the expression *S100A9* in this study. Thus, we speculated that *Muribaculaceae* may play a regulatory role in inflammation by interfering with the expression of *S100A9.* Additionally, we found another gene–bacteria pair that might serve the same purpose. *GADD45G* was enriched in NF-κ B and the MAPK signaling pathway was highly expressed in HSCC group. Though the genera (*Oscillospiraceae_UCG-005*) correlated with this gene have not been proven to be directly linked to subclinical mastitis before, their relative abundance did decrease in some other diseases [30,31]. 

With regard to the number of DEGs and differentially abundant genera, samples from LSCC were more easily discriminated from samples in MSCC. However, the number of differentially abundant genera in LSCC vs. HSCC comparison was smaller than in LSCC vs. MSCC. We conjectured that this was because some underlying factors may have unequal effects on certain milk microbiota derived from cows with different SCC levels, and the relationship between the abundance of some bacteria and SCC was not always linear [20]. 

To our knowledge, this is the first study to integrate host transcriptomics and microbiome to explore the cow-level SCC cut-off value for subclinical mastitis diagnosis. The results revealed a visible difference in microbiota profiling and gene expression pattern between composite samples with SCC above and below 100,000 cells/mL. Using 200,000 cells/mL as the threshold, however, led to a negligible difference in the intermediate molecular phenotypes, indicating 100,000 cells/mL can be more discriminative than 200,000 cells/mL. Though considerable effort has been devoted to finding the most reliable SCC threshold to predict mammary gland infection, it should be noted that the variability in SCC could be explained by different explanatory factors, and according to a previous study, sources of variability should be studied within and between animals before drawing a conclusion on a given animal’s health status [32]. As mentioned before, the current study was carried out based on composite milk samples. Quarter samples, which are more widely studied, on the other hand, are likely to obtain a different, maybe higher optimum mathematical SCC threshold as a result of the dilution effect caused by combining the milk for four udder quarters [9]. Thus, next time we may need to consider reconducting the study on the quarter level, further partitioning the variance based on the experimental factors and incorporating them into the model in order to better understand the role of each of the factors or the interplay between these factors in alternating the microbial composition and affecting the best cow-level and quarter-level SCC cut-off value for subclinical mastitis detection. 

## 5. Conclusions

In conclusion, our study pointed out the potential superiority of using 100,000 cells/mL as the cow-level threshold of subclinical mastitis detection over 200,000 cells/mL in terms of the changes in microbiota and transcriptome. Furthermore, the results showed that some beneficial bacteria may be helpful for alleviating subclinical mastitis progression. Thus, the differentially abundant genera and differentially expressed genes found in the current study might be conducive to the future diagnosis of intramammary infection. 

## Figures and Tables

**Figure 1 animals-12-01694-f001:**
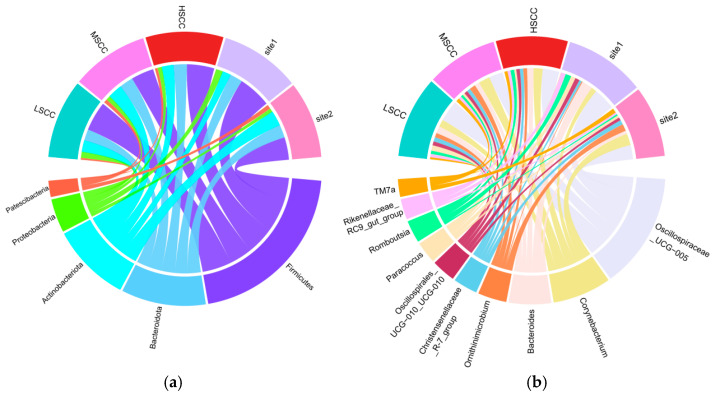
Microbiota composition at the phylum and genus level. (**a**): Top 5 most dominant phyla identified based on the mean relative abundance of all samples. (**b**): Top 10 most dominant genera identified based on the mean relative abundance of all samples.

**Figure 2 animals-12-01694-f002:**
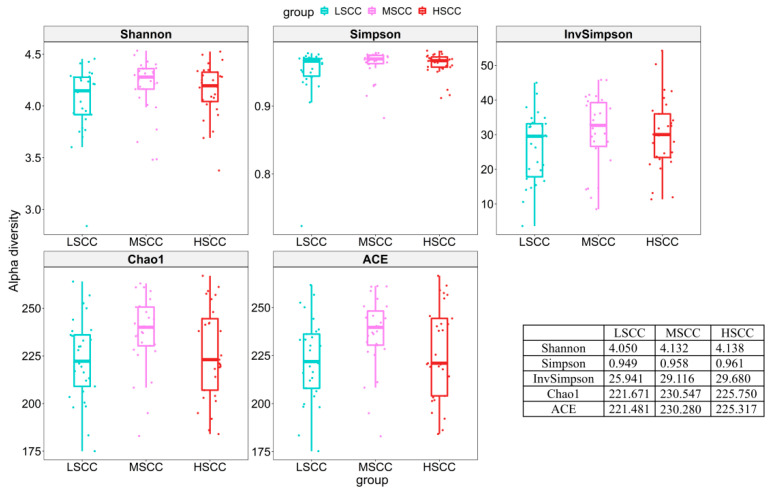
Boxplot showing different α diversity indices in each group. Estimated marginal means of each diversity index was assessed using mixed linear model.

**Figure 3 animals-12-01694-f003:**
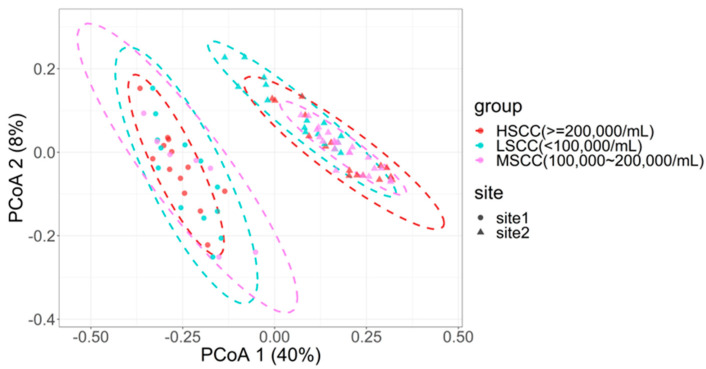
Principal coordinate analysis (PCoA) of Bray−Curtis distances among all milk samples.

**Figure 4 animals-12-01694-f004:**
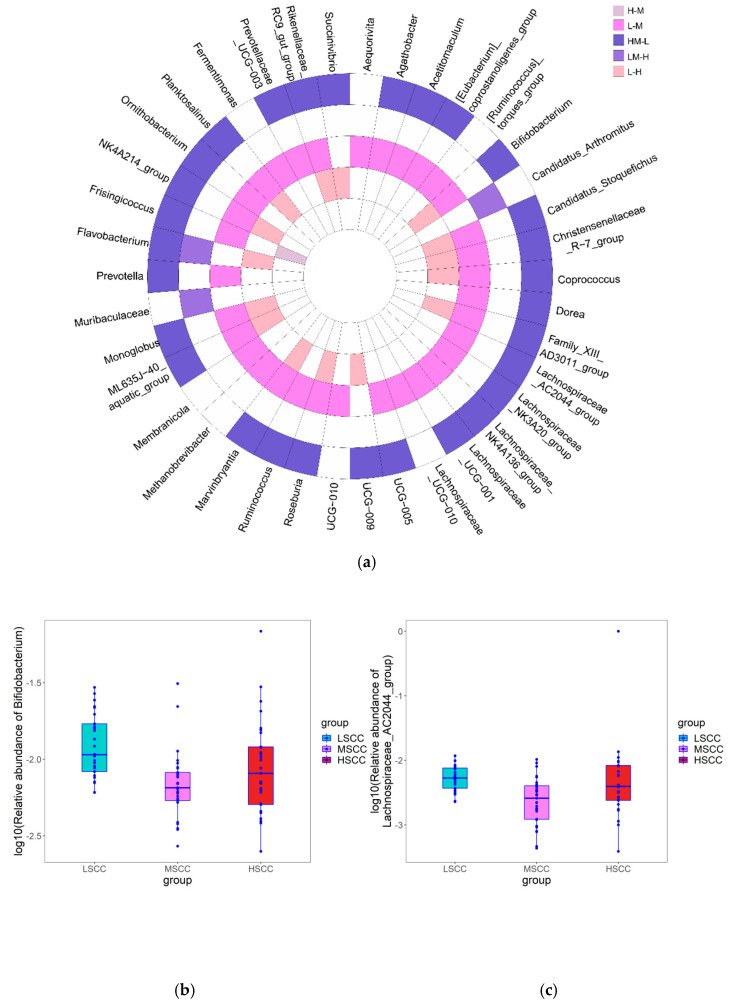
Differentially abundant genera and metabolic pathways obtained by pairwise comparison (**a**) Differentially abundant genera identified in each comparison. (**b**,**c**) Boxplot showing log10-transformed relative abundance of two most significantly differentially abundant genera between LSCC and MSCC (**d**) Venn diagram showing the number of the predicted metabolic pathways identified as significantly differentially abundant in each comparison and log10-transformed relative abundance of pathways that were shared in all comparisons. H−M: HSCC vs. MSCC, L−M: LSCC vs. MSCC, L−H: LSCC vs. HSCC.

**Figure 5 animals-12-01694-f005:**
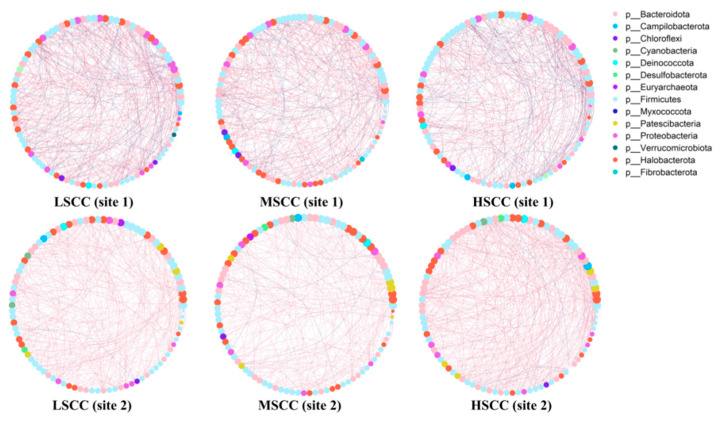
Co-occurrence networks of the top 100 predominant bacteria in different groups (spearman correlation > 0.8). Each node represents one genus. The size of the node represents mean relative abundance in the corresponding group. Red edges represent positive correlation and blue edges represent negative correlation.

**Figure 6 animals-12-01694-f006:**
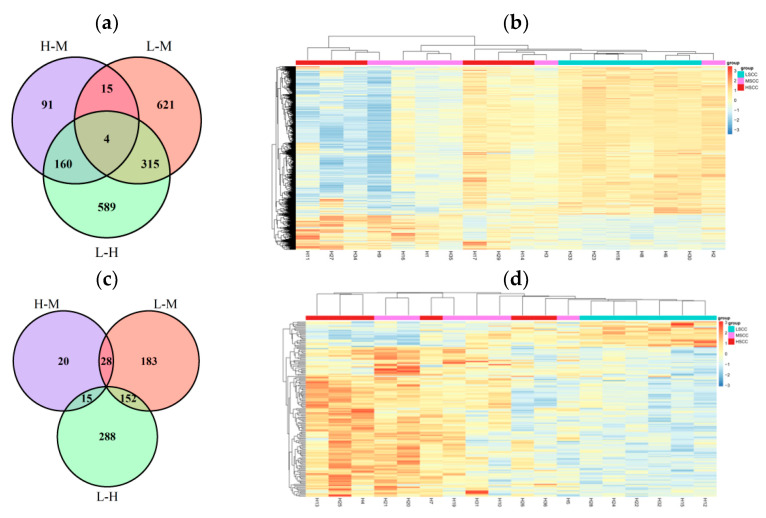
Differentially expressed genes identification. (**a**,**c**): Venn diagram showing the number of differentially expressed genes (DEGs) identified in each comparison in each site. (**b**,**d**): Heatmap showing the expression profile of top DEGs (FDR < 0.05). Left panel: site1, Right panel: site2. H−M: HSCC vs. MSCC, L−M: LSCC vs. MSCC, L−H: LSCC vs. HSCC.

**Figure 7 animals-12-01694-f007:**
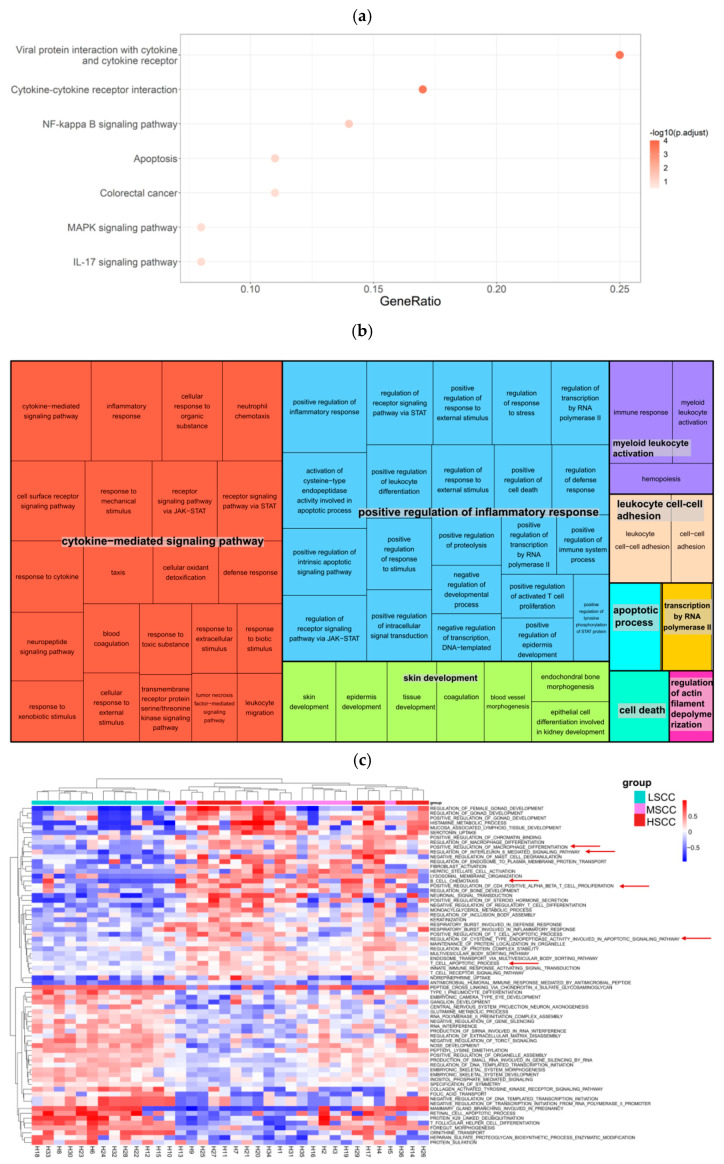
Gene functional analyses. (**a**,**b**) Bubble diagram and treemap illustrating the top 7 KEGG pathways and GO terms enriched by DEGs with the same regulation direction in two sites. (**c**) Heatmap showing the result of GSVA using biological process−related GO terms.

**Figure 8 animals-12-01694-f008:**
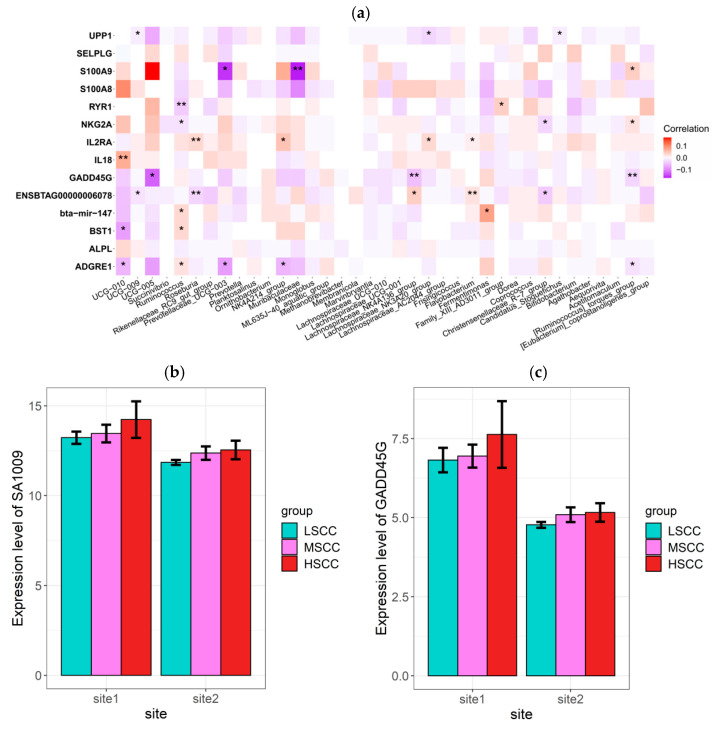
Correlation analysis between DEGs and differentially abundant taxa. (**a**) Associations between differentially abundant genera and the top discriminating DEGs identified in both sites using linear mixed model (‘*’stands for *p* < 0.05, ‘**’stands for *p* < 0.01). (**b**,**c**) Rlog-transformed read counts of *S100A9* and *GADD45G* in different groups.

**Table 1 animals-12-01694-t001:** Effect of different factors on milk microbial composition examined through Bray–Curtis distance-based redundancy analysis by group.

Factors	Groups Incorporated in the Model (F Statistic/*p* Value)				
	L-M (*n* = 58)	H-M (*n* = 58)	L-H (*n* = 58)	LM-H (*n* = 87)	HM-L (v = 87)
site	18.86/10^−4^	22.53/10^−4^	18.78/10^−4^	22.98/10^−4^	23.6/10^−4^
group	2.41/0.01	0.75/0.67	1.95/0.04	0.97/0.41	2.03/0.02
lactation stage	1.19/0.23	1.39/0.15	1.17/0.24	1.24/0.18	1.29/0.15
parity	1.19/0.22	1.10/0.28	1.22/0.21	1.23/0.18	1.17/0.21

Abbreviations: L = LSCC (low SCC; SCC < 100,000 cells/mL); M = MSCC (medium SCC; 100,000 ≤ SCC ≤ 200,000 cells/mL); H = HSCC (high SCC; SCC > 200,000 cells/mL); LM = LMSCC (low and medium SCC); HM = HMSCC (high and medium SCC).

**Table 2 animals-12-01694-t002:** Kolmogorov–Smirnov test of each microbial network topological parameter in two sites.

Comparison	Degree	Betweenness Centrality	Closeness Centrality	Transitivity
LSCC vs. HSCC	0.20 _a_ */0.19 _b_	0.08 _a_/0.19 _b_	0.90 _a_ **/0.99 _b_ **	0.19 _a_/0.25 _b_ **
LSCC vs. MSCC	0.13 _a_/0.17 _b_	0.15 _a_/0.08 _b_	0.98 _a_ **/0.79 _b_ **	0.08 _a_/0.32 _b_ **
MSCC vs. HSCC	0.14 _a_/0.36 _b_ **	0.10 _a_/0.18 _b_	0.98 _a_ **/0.99 _b_ **	0.19 _a_/0.42 _b_ **

* *p* ≤ 0.01, ** *p* ≤ 0.001, a: samples from site1, b: samples from site2.

## Data Availability

Not applicable.

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
