# Peer review of "Testing Two Somatic Cell Count Cutoff Values for Bovine Subclinical Mastitis Detection Based on Milk Microbiota and Peripheral Blood Leukocyte Transcriptome Profile"

_animals, 2022, doi:10.3390/ani12131694_

Round 1

Reviewer 1 Report

Introduction

The authors do not really seem to understand the difference between subclinical mastitis threshold for individual animals and for herds. That way they really confuse the readers. The first paragraph should be really rewritten correctly. Based on the rewritten para, they should reformulate their hypothesis on better grounds really…..

Threshold 100,000 can be correct at animal level and threshold 200,000 can also be very acceptable at herd level. The authors do not seem to distinguish the differences and that way they create further confusion. Not mentioning that their hypothesis is not solid.

M & M

2.1. How many cows? What criteria for selection and inclusion?

2.2. All the details for the PCRs in a separate table in supplementary material please.

Results

These are OK, BUT they are within the wrong context.

Please correct context of the manuscript from start and then these result can become valuable.

The study needs significant revision before acceptance. The authors did not present the work within the right context. The discrimination between two different type of thresholds is misleading and is bad for the manuscript.

Significant revision as proposed above and re-evaluation.

Reviewer 2 Report

This study is the first that reports on the microbiome of milk samples from cows with different levels of somatic cell count. Authors used acceptable technology and analysis.  All my comments are included in the annoted PDF.

I sincerely hope these commets are accepted as a constrictive cirticism only.

Round 2

Reviewer 1 Report

The authors mention the collection of 87 milk, but these originated from how many cows? This is not mentioned clearly.

Author Response

Thanks for your review and feedback. These 87 samples were originated from a total of 121 cows as mentioned in M & M. The original text is as follows: A total of 121 milk samples were divided into the following three groups based on SCC value.

Reviewer 2 Report

Authors have addressed all my comments satisfactorily.  Minor changes required (in the attached file)

Author Response

Thanks for your kindly suggestions. All of the improper words or phrases we used have been replaced according to your advice. The details of revisions can be found in the revised manuscript.